



**Physicochemical and Temporal Characteristics of Individual**
**Atmospheric Aerosol Particles in Urban Seoul during KORUS-AQ**
**Campaign: Insights from Single-Particle Analysis**
Hanjin Yoo[1,2], Li Wu[3], Hong Geng[4,+], and Chul-Un Ro[1,2,+]
[1]Department of Chemistry, Inha University, Incheon, 22212, Republic of Korea
[2]Particle Pollution Management Center, Inha University, Incheon, 21999, Republic of Korea
[3]School of Earth Science System, Tianjin University, Tianjin, China
[4]Institute of Environmental Science, Shanxi University, Taiyuan, China
*Correspondence to:* Chul-Un Ro (curo@inha.ac.kr) and Hong Geng (genghong@sxu.edu.cn)
**ABSTRACT**

15       Single-particle analysis was conducted to characterize atmospheric aerosol particles

collected at Olympic Park in Seoul, Korea as a part of the KORUS-AQ campaign which was
carried out during May-June 2016. The KORUS-AQ campaign aimed to understand the
temporal and spatial characteristics of atmospheric pollution on the Korean Peninsula through
an international cooperative field study. A total of 8004 individual particles from 52 samples
collected between 5/23-6/5, 2016, were investigated using a quantitative electron probe X-ray
microanalysis (low-*Z* particle EPMA), resulting in the identification of seven major particle
types. These included genuine and reacted mineral dust, sea-spray aerosols, secondary aerosol
particles, heavy metal-containing particles, combustion particles, Fe-rich particles, and others
(biogenic and humic-like substances (HULIS) particles). Distinctly different relative
abundances of individual particle types were observed during five characteristic atmospheric
situations, namely (a) a mild haze event influenced by local emissions and air mass stagnation,
(b) a typical haze event affected by northwestern air masses with a high proportion of sulfate-
containing particles, (c) a haze event with a combined influence of northwestern air masses and
local emissions, (d) a clean period with low particulate matter concentrations and a blocking
pattern, and (e) an event with an enhanced level of heavy metal-containing particles, with Zn,
Mn, Ba, Cu, and Pb being the major species identified. Zn-containing particles were mostly
released from local sources such as vehicle exhausts and waste incinerations, while Mn, Ba,
and Cu-containing particles were attributed to metal-alloy plants or mining. The results suggest



that the morphology and chemical compositions of atmospheric aerosol particles in urban area
vary depending on their size, sources, and reaction or ageing status, and are affected by both
local emissions and long-range air masses.

Key Words: KORUS-AQ Campaign; low-*Z* particle EPMA; urban megacity; haze

**Introduction**
Atmospheric aerosols, originating from various anthropogenic and natural sources,
have significant impacts on climate change and human health (**IPCC, 2021**). Anthropogenic
emissions greatly influence the composition and behavior of airborne particulate matter (PM)
(**Kim et al., 2018b; Chowdhury et al., 2018; Nault et al., 2018**). On the Korean Peninsula,
anthropogenic pollutants primarily come from local emissions and long-range transport air
masses (**Cho et al., 2021; Park et al., 2020; Choi et al., 2021; Kumar et al., 2021; Nault et**
**al., 2018**). Studies have observed changes in the characteristics of aerosols composed of
organic and inorganic compounds influenced by different air mass flows. Secondary organic
aerosols (SOAs) are particularly affected by local emissions, while inorganic particles can be
influenced by either local emissions or long-range transported pollutants (**Nault et al., 2018;**
**Kim et al., 2018b; Kim et al., 2020; Park et al., 2018; Chen et al., 2017**). Local emissions,
including biomass burning, cooking, and traffic exhaust, primarily influence the formation of
SOAs in urban areas (**Nault et al., 2018; Park et al., 2018; Kim et al., 2018b**). On the other
hand, transboundary transport of pollutants is significantly affected by comprehensive climatic
conditions and can lead to air pollution episodes dominated by inorganic components,
including sulfate, nitrate, and ammonium (**Kumar et al., 2021; Lee et al., 2019a; Choi et al.,**
**2019**).
East Asia has seen a significant decline in air quality over the past few decades due to
increased emissions of gaseous and particulate pollutants as a result of rapid industrial and
economic growth. The Korean Peninsula, surrounded by China, Japan, and Russia, exhibits
complex aerosol characteristics influenced by a combination of local emissions, surrounding
seas, and transboundary long-range transported air masses (**Pochanart et al., 2004; Crawford**
**et al., 2020; Peterson et al., 2019; Ramachandran et al., 2020; Kim et al., 2018a**). To further
investigate factors affecting air pollution on the Korean Peninsula, an international cooperative
field study, the KORUS-AQ (Korea-US Air Quality) campaign, was conducted during May-
June 2016 (**Crawford et al., 2020**). Through this campaign, the temporal and spatial
characteristics of various gaseous and particulate pollutants on the Korean Peninsula were



successfully elucidated, making it an important study in the field of atmospheric science
(**Crawford et al., 2020**). In the Korean Peninsula, ammonium was found to be the most
sensitive factor affecting $PM_{2.5}$ exposure, followed by $NO_x$, $SO_2$, organic carbon (OC), and
black carbon (BC) (**Choi et al., 2019**). The presence of anthropogenic ammonium on the
Korean Peninsula leads to the formation of ammonium sulfate (AS) and ammonium nitrate
(AN) particles (**Kim et al., 2021; Kim et al., 2020**). Regarding the composition of atmospheric
$PM_1$ in Seoul, the most populated metropolitan area in Korea, OC content was found to be the
highest, followed by sulfate, nitrate, ammonium, and BC (**Kim et al., 2018b**).

While previous studies have effectively examined the impact of anthropogenic
emissions on the formation of submicron particles during the KORUS-AQ campaign, research
on supermicron particles remains limited. Aerosol particles in the supermicron fraction, which
mainly originate from natural sources like mineral dust and sea-spray aerosols (SSAs), make
up a significant proportion of the total aerosol mass (**Andreae and Rosenfeld, 2008; Seinfeld**
**and Pandis, 2006**). Airborne mineral dust particles in East Asia, directly emitted from arid
regions of Mongolia and northern China, can undergo physicochemical changes during long-
range transportation, for example, through atmospheric reactions with anthropogenic $NO_x$ and
$SO_2$, resulting in the formation of nitrates and sulfates. This leads to alterations in chemical
compositions, morphology, size, and radiative forcing capabilities (**Sullivan et al., 2007; Yu**
**et al., 2020; Geng et al., 2014; Heim et al., 2020; Sobanska et al., 2012**). The investigation
of the characteristics of supermicron particles, including their particle-particle variability,
formation dynamics, and atmospheric fate, is important to gain a comprehensive understanding
of the behavior and impact of atmospheric aerosols of natural and anthropogenic origin on air
quality and climate change.

This study utilized a quantitative electron probe X-ray microanalysis (EPMA)
technique based on scanning electron microscopy coupled with X-ray spectrometry, so-called
low-$Z$ particle EPMA, to examine the physicochemical characteristics of individual aerosol
particles collected at Olympic Park in Seoul, Korea during the KORUS-AQ campaign. Low-$Z$
particle EPMA is a powerful single-particle analytical technique for providing information on
unique features of individual aerosol particles, including morphology, elemental compositions,
and particle-particle variability (**Geng et al., 2009; Geng et al., 2011; Li et al., 2017; Wu et**
**al., 2019**). Differences in these features are attributed to particle sources, formation
mechanisms, and atmospheric fate (**Wu et al., 2019; Song et al., 2022**). This article consists
of two parts: (1) an examination of the differences in physicochemical characteristics based on
particle species and (2) an analysis of the temporal variations of individual aerosol particles



during the KORUS-AQ campaign. The characterization of individual particles, combined with
other studies on atmospheric aerosols during the KORUS-AQ period, provides valuable
insights into the unique features of urban atmospheric particles.

**2. Experiments**
**2.1 Sampling**

Ambient aerosol particles were collected at Olympic Park (37.52° N, 127.12° E) in

Seoul, the capital of South Korea (Fig. S1 in Supporting Information). The Seoul metropolitan
area (SMA), with high population density, numerous local emissions, and transboundary long-
range transport, provides a suitable location for investigating the complex characteristics of
atmospheric aerosols (**Kim et al., 2018b**; **Kim et al., 2020**). A 3-stage cascade Dekati PM10
impactor (Dekati Ltd.) with an aerodynamic cut-off size of 10, 2.5, and 1.0 μm for stages 1-3
at a 10 L min$^{-1}$ flow rate, respectively, was used to collect aerosol particles on Al foils. Each
sample set was analyzed for particles collected on stages 2 and 3, corresponding to $PM_{2.5-10}$
and $PM_{1-2.5}$, respectively. A total of 52 sets of samples were collected in the morning and
evening (9:00 ~ 10:00 and 15:00 ~ 16:00, KST) during May 23 to June 5, 2016. The sampling
duration for each stage was controlled to obtain an optimum number of particles without
overloading on the Al foils. 72-hour backward air mass trajectories were generated using the
HYSPLIT (Hybrid Single-Particle Lagrangian Integrated Trajectory) model for different
receptor heights of 250 m, 500 m, and 1000 m above ground level. The HYSPLIT model is
available    at    the    NOAA    Air    Resources    Laboratory's    website
(http://www.arl.noaa.gov/ready/hysplit4.html).

**2.2 Determination of individual particle types by low-$Z$ particle EPMA**

The physicochemical characteristics of individual aerosol particles were examined

using a SEM (JEOL JSM-6390) equipped with an Oxford Link SATW ultrathin window EDX
detector. The resolution of the detector was 133 eV for Mn-Kα X-rays, and X-ray spectra were
recorded using INCA Oxford software (Oxford Instruments Analytical Ltd, INCA suite version
4.09). Routine measurement was conducted using an accelerating voltage of 10 kV and beam
current of 0.5 nA, while 20 kV and 0.25 nA were used to confirm heavy metal elements of
specific particles. To obtain sufficient X-ray counts for quantitative analysis, a typical
measurement time of 15 s was chosen. The net X-ray intensities for the elements were obtained



using a non-linear least-squares fitting of the collected spectra using the AXIL program
(**Vekemans et al., 1994**). The elemental concentrations of the individual particles were
determined from their X-ray intensities using a Monte Carlo calculation combined with reverse
successive approximations (**Ro et al., 2001, 2002**). The chemical species of individual aerosol
particles were determined based on their size, morphology, and elemental composition.

**3 Results and discussion**
**3.1 Characteristics and abundances of individual particle types**
Individual particles were classified into 13 species based on their morphology and
elemental composition, and further categorized into seven major groups based on their sources
and/or formation mechanism. These groups are (1) secondary aerosol particles including SOAs
and secondary organic and inorganic aerosols (SOIAs), (2) genuine and aged/reacted mineral
dust, (3) reacted SSAs, (4) combustion particles, (5) Fe-rich particles, (6) heavy metal-
containing particles, and (7) others, including biogenic and humic-like substances (HULIS)
particles. More information on the classification can be found in the Supporting Information
(Section A and Tables S1).

*3.1.1 Secondary aerosol particles (SOAs and SOIAs)*
Secondary aerosol particles, including SOAs and SOIAs, account for 5.6% and 29.3%
in the $PM_{2.5-10}$ and $PM_{1-2.5}$ fractions, respectively. These particles, likely formed through gas-
to-particle conversion, photochemical processes, and the condensation of semi-volatile organic
compounds (**Hallquist et al., 2009; Kim et al., 2018a**), are significantly more abundant in the
fine $PM_{1-2.5}$ fraction than in the $PM_{2.5-10}$ fraction. The morphology, X-ray spectra, and elemental
compositions of typical secondary aerosol particles are presented in Fig. 1. SOA particles
appear as dark droplets in their secondary electron image (SEI) and are primarily composed of
C and O (>90% in low-*Z* particle EPMA analysis) (Fig. 1a). The spread droplet-like
morphology of SOA particles collected on the hydrophilic Al foil suggests that they are likely
low-viscous and water-soluble. In contrast, SOIAs, which are mixtures of SOA and inorganic
constituents such as $NH_4^+$, $NO_3^-$, and/or $SO_4^{2-}$, exhibit C, N, O, and S in their X-ray spectra and
are apparently susceptible to damage by electron beams (Figs. 1b and 1c). The morphology of
SOIA particles varies depending on the organic and inorganic contents. Those with high
inorganic content appear as bright, crystalline shapes surrounded by a water-soluble footprint
(Fig. 1b), while those with a high organic content resemble dark droplets (Fig. 1c). An inset



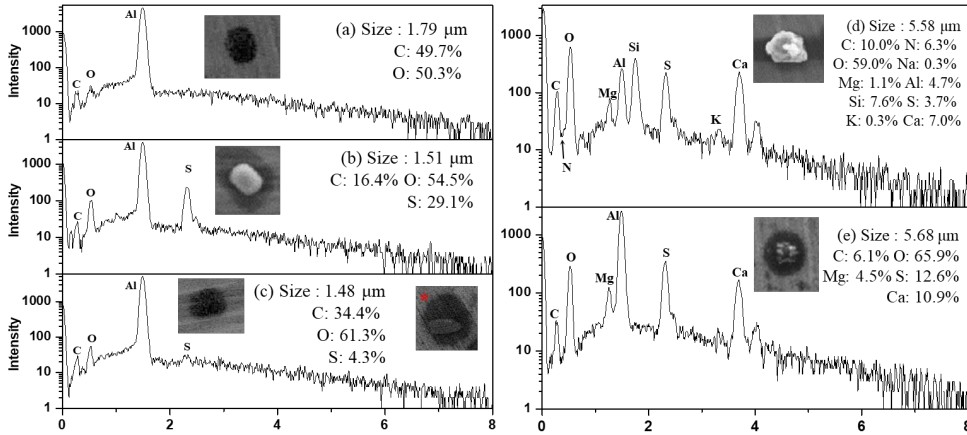

**Figure 1. Morphology, X-ray spectra, and elemental compositions of (a) SOA, (b) SOIA (high inorganic), (c) SOIA (high organic), (d) reacted aluminosilicate, and (e) reacted carbonate particles.**


marked with * on Fig. 1c shows a SOIA particle that appears as a core-shell structure, with a
SOIA core surrounded by a dark droplet shade mainly containing C and O. The differences in
the crystalline morphology of SOIAs indicate that the heterogeneous nucleation and/or
crystallization of particles can vary depending on the chemical species present (**Wu et al.,**
**2020**). Furthermore, the significant water-soluble footprint surrounding SOA and SOIA
particles indicates that aqueous-phase chemistry is a crucial process in the formation of
secondary aerosol particles in the urban area of Seoul. Previous studies have reported that SOA
particles in South Korea are primarily influenced by local emissions, while the sources of
inorganic components are highly relevant to both local emissions and transboundary long-range
transport air masses (**Nault et al., 2018; Kim et al., 2018b; Choi et al., 2019**).
*3.1.2 Mineral dust particles*
Genuine and reacted mineral dust particles are the most abundant particle types among
the seven major ones in this study, accounting for 73.2% and 44.5% in the $PM_{2.5-10}$ and $PM_{1-2.5}$
fractions, respectively. These mineral dust particles are irregularly shaped and appear bright
in SEI, mainly consisting of crustal elements such as Al, Si, Ca, Mg, K, and others. The
observed chemical species of mineral dust particles include aluminosilicates (such as feldspar,
muscovite, montmorillonite, illite, kaolinite, talc, pyrophyllite, etc.), quartz ($SiO_2$), carbonates
(calcite ($CaCO_3$), dolomite ($CaMg(CO_3)_2$), and magnesite ($MgCO_3$)), $TiO_2$, and their
reacted/aged ones. Genuine mineral dust particles, tending to be larger in size, are significantly



more abundant in the PM$_{2.5-10}$ fraction than in the PM$_{1-2.5}$ fraction, whereas the proportion of
reacted minerals is slightly higher in the PM$_{1-2.5}$ fraction (71.0%) compared to the PM$_{2.5-10}$
(66.2%) due to the larger specific surface area of PM$_{1-2.5}$ particles, making them more prone to
chemical reactions in the air.
The reactivity of mineral dust particles varies depending on their chemical species and
size, as shown in Table 1. In the PM$_{2.5-10}$ fractions, particles are highly associated with nitrate
compared to sulfate (46.2% vs. 30.0%), while the abundance of sulfate is comparatively higher
than nitrate for PM$_{1-2.5}$ particles (34.3% vs. 20.0%), indicating that sulfate formation occurs
more frequently in smaller particles. The proportion of reacted particles is significantly higher
in carbonate particles than in aluminosilicates (93.9% vs. 56.2%), indicating that carbonate
mineral dust has a higher reactivity than aluminosilicates. Reacted aluminosilicate particles
appear bright and irregular, being surrounded by water-soluble moieties (Fig. 1d), implying
that the chemical reaction mostly occurred on the surface, while reacted carbonate species show
dark lumpy, core-shell shapes (Fig. 1e), indicating that the reaction readily occurred from the
surface to the internal part. Further analysis reveals that the carbonate particles tend to react
with sulfate, while aluminosilicates were more likely to interact with nitrate (Table 1). The
different abundances of sulfate and nitrate in the reacted mineral particles not only depend on
the particle species and size, but also on the source, transport pathway, and formation process
(**Geng et al., 2011, 2014; Sullivan et al., 2007**). These findings suggest that (a) carbonate
minerals are more sensitive to changes in atmospheric conditions than aluminosilicates, and (b)
carbonate minerals react with sulfate before nitrate due to the prevailing neutralization by
sulfate (**Takahashi et al., 2014; Matsuki et al., 2005; Seinfeld and Pandis., 2006; Sullivan**
**et al., 2017**).

**Table 1. Relative abundances of genuine and reacted mineral dust and SSA particles**

| Type | | Genuine | Reacted | | | % of reacted particles |
|---|---|---|---|---|---|---|
| | | | Containing-N | Containing-S | Containing-both | |
| **PM$_{2.5-10}$** | | | | | | |
| Mineral dust | Aluminosilicates | 23.2% | 17.9% | 4.9% | 5.8% | 55.2% |
| | Carbonates | 1.5% | 8.5% | 6.9% | 4.5% | 92.9% |
| Sea spray aerosols | | | 4.5% | 2.9% | 5.1% | 100% |
| **PM$_{1-2.5}$** | | | | | | |
| Mineral dust | Aluminosilicates | 12.1% | 5.0% | 7.7% | 3.4% | 57.2% |
| | Carbonates | 0.8% | 3.6% | 10.2% | 1.6% | 95.0% |
| Sea spray aerosols | | | 4.4% | 9.4% | 2.0% | 100% |




*3.1.3 Sea-spray aerosols (SSAs)*
Nascent SSAs are rich in characteristic elements such as Na, Mg, and Cl, as indicated
by their X-ray spectra. They are released into the atmosphere from the sea surface through film
drops and jet drops caused by bubble bursting (**Eom et al., 2016; Cochran et al., 2017**).
Freshly emitted SSAs are a mixture of inorganic Na, Mg, and Cl and organic compounds such
as fatty acids, amino acids, and liposaccharides, which are closely related to the biological
activity of micro-organisms in the marine environment (**Eom et al., 2016; Cochran et al.,**
**2017**). Once released into the atmosphere, these nascent SSAs tend to react with various acidic
species such as sulfuric, nitric, and organic acids to form reacted/aged SSAs. All SSAs for both
$PM_{2.5-10}$ and $PM_{1-2.5}$ fractions were found to be in the reacted form (Table 1), despite their short
transport distances (~50–100 km until they reach the sampling site from the Yellow Sea),
suggesting that they are susceptible to atmospheric reactions (**Laskin et al., 2003; Gupta et**
**al., 2015; Li et al., 2017; Chen et al., 2020).** As shown in Table 1, the reacted SSAs accounted
for 12.4% and 15.7% in the $PM_{2.5-10}$ and $PM_{1-2.5}$ fractions, respectively, in which the nitrate-
containing SSAs were more abundant than the sulfate-containing ones in the $PM_{2.5-10}$ fraction
(9.6% vs. 8.4%), while those containing sulfates were more abundant in the $PM_{1-2.5}$ fraction
(11.3 vs. 6.4%), indicating that sulfate formation occurs more in smaller SSA particles. The
higher abundance of SSAs containing both nitrates and sulfates in the larger size fraction may
be attributed to the availability of sufficient anions to accumulate acidic cations, which is
associated with a decrease in acidity as particle size increases (**Angle et al., 2021).**
*3.1.4 Combustion particles*
The combustion particles include soot agglomerates, tar balls, fly ash, and char particles,
accounting for 1.3% and 2.8% in the $PM_{2.5-10}$ and $PM_{1-2.5}$ fractions, respectively. Most
elemental carbon (EC) particles, such as soot agglomerates, tar balls, and char particles, have
similar elemental compositions, but they can be differentiated based on their unique
morphology (Fig. 2 and Table S1).
Soot agglomerates are remnants of incomplete combustion and are formed through the
vaporization-condensation mechanism (**Bond et al., 2004; Chen et al., 2006**). Based on their
morphology and elemental compositions, soot agglomerates can be classified into two types:
fresh and aged. The fresh soot agglomerates appear bright and have a characteristic chain-like
structure with fractal geometry, as shown in the right-side SEI of Fig. 2a. The complex
geometry of the soot agglomerates provides an active area for the deposition of gaseous or



particulate species. The morphology of aged soot agglomerates shown in Fig. 2a is more
compact than that of the fresh ones. The aging of the soot agglomerates is attributed to several
mechanisms such as oxidation, absorption or condensation of gaseous species, and coagulation
with other particles. This aging process can cause the soot agglomerates to shrink and
restructure into a more compact shape, as shown in Fig. 2a **(Bond et al, 2004; Zhang et al.,**
**2008; Chen et al., 2006)**.

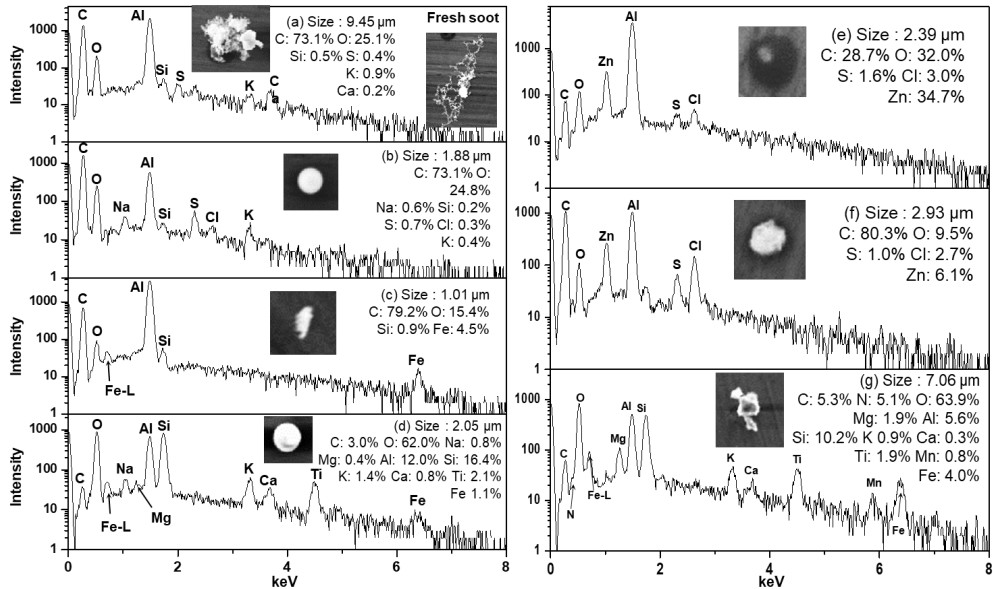

**Figure 2. Morphology, X-ray spectra, and elemental compositions of (a) aged soot aggregates, (b) tar balls, (c) char, (d) fly ash, (e) and (f) Zn-HMs, and (g) Mn/Ti-HMs.**

Tar ball particles are composed of organic oligomers and are a representative particle
type from smoldering combustions such as biomass burning or biofuel combustion (**Adachi et**
**al., 2019; Girotto et al., 2018; Pósfai et al., 2004**). The spherical shape of the tar ball particles
(Fig. 2b) results from post-physical and chemical transformation of the organic matter. The
formation of the tar balls can vary depending on factors such as oligomerization of organics,
condensation, photochemical processes, water loss, and temperature changes, leading to
different internal structures (**Tóth et al., 2018; Adachi et al., 2019**).
Char particles are incomplete combustion residues of liquid or solid carbonaceous fuel
materials, appearing compact and irregular in shape on the SEI, as shown in Fig. 2c (**Chen et**
**al., 2006**).





Fly ash particles, as shown in Fig. 2d, have a similar elemental composition to
aluminosilicate mineral particles but with a distinct bright spherical shape on the SEI. These
particles were rarely found in both size fractions, accounting for 0.08% and 0.42% in $PM_{2.5-10}$
and $PM_{1-2.5}$ fractions, respectively. The spherical morphology of fly ash particles is attributed
to their formation mechanism, which involves rapid cooling after being released from high-
temperature combustion at industrial plants **(Geng et al., 2011).**

*3.1.5 Heavy metal-containing particles (HMs)*
Particles containing heavy metal elements (HMs), such as Zn, Pb, Cu, Mn, Ba, Zr, Sr,
Cd, As, Cr, V, Ni, Sn, and Co, are of particular concern due to their adverse impact on human
health. In this study, a significant number of HMs were observed, accounting for 2.7% and 4.4%
in the $PM_{2.5-10}$ and $PM_{1-2.5}$ fractions, respectively. Among the 14 types of HMs observed, Zn,
Pb, Ba, Cu, and Mn were frequently encountered (Fig. 3). HMs can be released from both
anthropogenic and natural sources, with thermal power plants, vehicle exhaust, battery
manufacture, and the metallurgical industry being some of the most common anthropogenic
sources (**Tian et al., 2015; Xu et al., 2004**). Tracing the sources of HMs during the KORUS-
AQ campaign can be done based on coexisting elements, morphologies, and relative
abundances.


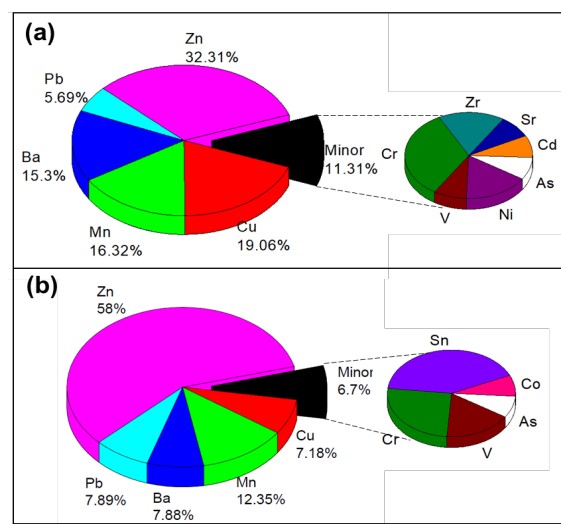

**Figure 3. Heavy metals observed in HMs for (a)
$PM_{2.5-10}$ and (b) $PM_{1-2.5}$ fractions.**



As shown in Fig. 3, the most abundant type of HMs observed in this study were Zn-
containing particles (Zn-HMs), accounting for 32.3% and 58.0% of the total HMs in $PM_{2.5-10}$
and $PM_{1-2.5}$, respectively. Zn-HMs can be emitted from various anthropogenic sources such as
waste incineration, vehicle emissions, rubber tire wear, and coal combustion (**Hopke et al.,**
**1991; Chow et al., 2004; Hjortenkrans et al., 2007**). In the sampling site, which is an urban
area with heavy traffic, Zn-HMs may be attributed to vehicle emissions such as rubber tire and
brake pad wear. Two major identified types of Zn-HMs were C-Zn-Cl and C-Zn-Cl + (N or S)
(Fig. 2e and 2f), which made up 54.4% and 29.8% of the total Zn-HMs, respectively. A
significant proportion (84.9%) of Zn-HMs were observed to contain Cl, likely due to
incomplete atmospheric reactions of $ZnCl_2$. $ZnCl_2$ can easily undergo aqueous-phase chemical
reactions in the atmosphere due to its hygroscopic nature. The presence of N or S on the X-ray
spectra and dark droplet morphology on the SEI of the Zn-HMs indicate that the particles had
undergone atmospheric reactions with $NO_x/SO_x$ (**Moffet et al., 2008**). The temporal variations
of Zn-HMs will be discussed in Section 3.2.
A total of 20 Pb-HMs were observed in this study, in the forms of mixtures with SSAs
(8 particles), Pb-Cl-other heavy metals (6 particles), mineral dust (4 particles), and Pb-As (2
particles). They were likely emitted from vehicle exhaust and coal-fired power plants (**Lee et**
**al., 2019b**). Among the 39 Mn-HMs observed, 24 particles were associated with mineral dust,
coexisting mainly with Al, Si, Ca, and Mg; 6 particles with Fe; 2 particles with SSAs; 4
particles with Mg, Cl, and S; and 3 particles with F. They might originate from natural soil or
anthropogenic sources such as ore-crushing plants, ferroalloy plants, and similar facilities
(**Moreno et al., 2011**). The morphology and elemental composition of a Mn-HM are shown in
Fig. 2g. Among the total 33 Cu-HMs, 17 particles were mixed with Fe, followed by the mineral
dust form (10 particles), and minor forms such as Cu-C-S and Cu-C-N-O (6 particles). Major
sources of atmospheric Cu include non-ferrous metal plants, mining, and smelting complexes
(**Choi et al., 2013; Eichler et al., 2014**). Among the total of 30 Ba-HMs, 17 particles were
mixed with Fe, followed by mineral dust (5 particles), $BaSO_4$ (3 particles), and other minor
forms (5 particles). Ba-HMs could be released from natural sources in the form of barite
($BaSO_4$) and witherite ($BaCO_3$), and anthropogenic sources such as ore crushing plants, mining,
refining, and manufacture of barium products (**Choudhury et al., 2009; Beddows et al., 2004**).
The fact that Mn, Ba, and Cu-HMs appear abundantly as a mixture of Fe or mineral dust
suggests that their major source might be ferroalloy plants, mining, or ore crushing plants.
*3.1.6 Fe-rich, biogenic, and HULIS particles*



Fe-rich particles, which have an irregular shape and appear bright on the SEI, usually
contain more than 20% Fe in elemental concentration. These particles account for 1.7% and
2.2% in the $PM_{2.5-10}$ and $PM_{1-2.5}$ fractions, respectively, and likely originate from steel
production, metallurgical industries, and the abrasion of brake linings (**Geng et al., 2011**).
Biogenic particles, primarily originating from natural sources (**Martin et al., 2010**), are
relatively more abundant in the $PM_{2.5-10}$ fraction (2.83%) than the $PM_{1-2.5}$ fraction (0.81%).
They can be identified by their unique morphologies and the presence of minor elements such
as Na, Mg, N, K, P, S, and/or Cl (**Ro et al., 2002; Geng et al., 2011**). In this study, most of the
observed biogenic particles were attributed to trichomes, plant fragments, pollen, or spores, as
their sizes were generally larger than 2 μm (**Matthias-Maser et al., 2000; Coz et al., 2010**).
Typical examples of biogenic particles are displayed in Fig. S2a-c, corresponding to fungal
spores, micro-organism, and trichomes or leaf fragments, respectively.
The HULIS particles, consisting mainly of water-insoluble organic carbon (WISOC),
are characterized by high C and O content and unique morphology. There are 17 out of 8004
particles, only accounting for 0.2%. They might be released from soil, wetland, and sewage-
treatment plants.

**3.2 Temporal chemical composition variations of individual aerosol particles during the**
**KORUS-AQ campaign**
Based on differences in relative abundances of individual particle types, ambient PM
concentrations (Fig. S3), and backward air mass trajectories (Fig. S4), the sampling period
(5/23–6/5) of the KORUS-AQ campaign was divided into five characteristic atmospheric
situations as follows: (Period I, 5/23) - a SOA dominant period influenced by local emissions
and air mass stagnation; (Period II, 5/25-5/28) - a SOIA-rich haze episode with the influence
of long-range transported air-masses; (Period III, 5/29-5/31) - haze events with the combined
influence of long-range transported air-masses and local emissions; (Period IV, 6/1-6/3) - a
clean air period; and (Period V, 6/4-6/5) - a period dominantly influenced by local emissions.
The relative abundances of individual particle types are shown in Fig. 4. There are significant
differences over the sampling period, especially in the $PM_{1-2.5}$ fractions.



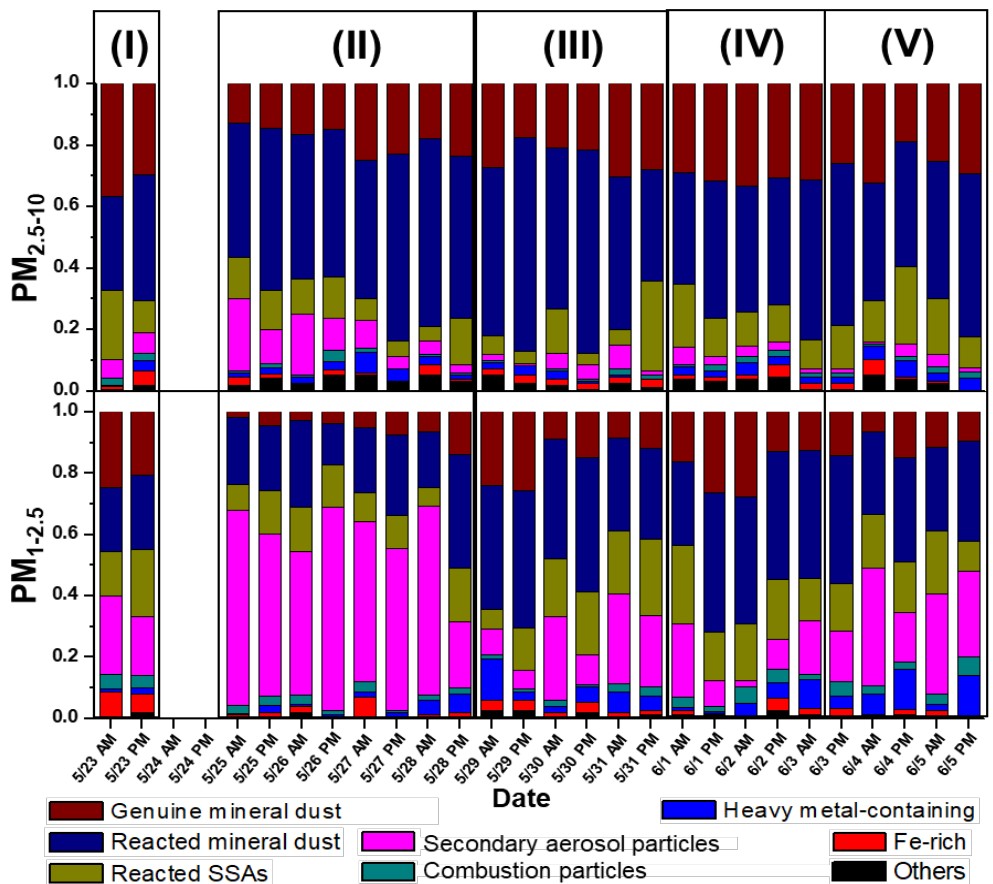

**Figure 4. Relative abundances of various particle types in the PM$_{2.5-10}$ and PM$_{1-2.5}$ fractions.**


**Period I (5/23)**

On 5/23, the first day of the sampling period, individual aerosol particles showed clear

distinctions in morphology and elemental compositions, particularly for secondary aerosol
particles. As shown in Fig. 5a, most secondary aerosol particles in the PM$_{1-2.5}$ fraction,
including SOA and SOIA, had dark droplet morphology, indicating that their major chemical
species are organic carbon. Figure 6 highlights a significant increase in the ratio of SOA to
secondary aerosol particles. The SOA/secondary aerosol particles ratio of the PM$_{1-2.5}$ fraction
was notably higher (55.2%) in the 5/23 sample compared to the average for the overall samples
during the campaign (22.0%), emphasizing the enhanced contribution of organic carbon to
secondary aerosol particle formation. Figure 7 shows that the proportion of combustion





particles increased by 1.8 and 1.6 times compared to the overall average in the $PM_{2.5-10}$ and
$PM_{1-2.5}$ fractions, respectively. A slightly elevated PM concentration on 5/23 (Fig. S3) suggests
mild air pollution on that day. Our findings align with other bulk studies that confirmed an
increased proportion of organic carbon in $PM_1$ aerosols during 5/17-5/23 (**Kim et al., 2018a;**
**Kim et al., 2018b**). Stagnant conditions under a persistent anticyclone prevented the transport
of pollutants from other regions, suggesting a dominant influence of local emissions during
this period (**Kim et al., 2018b; Peterson et al., 2019; Heim et al., 2020**). The formation of
SOA in South Korea, particularly in urban areas, was reported to be predominantly influenced
by local emissions (**Nault et al., 2018**). Consequently, the rise in the proportion of organic
carbon during Period I can be attributed to the augmented contribution of local emissions to
the formation of secondary aerosol particles due to air mass stagnation (**Peterson et al., 2019;**
**Kim et al., 2018a; Kim et al., 2018b; Crawford et al., 2021**). The enhanced level of
combustion particles also suggests the contribution of local emissions. Overall, the data from
5/23 indicate a clear influence of local emissions on aerosol particle composition and
concentration.

**Period II (5/25-5/28)**

After the rainfall on 5/24, the morphology, elemental composition, and relative

abundance of individual aerosol particles during 5/25-5/28 (Period II) differed significantly
compared to those observed in Period I. In terms of particle morphology, Fig. 5b shows that
SOIA particles on 5/25 exhibited a bright crystalline morphology, suggesting that these
particles are primarily composed of inorganic components such as sulfate, nitrate, and

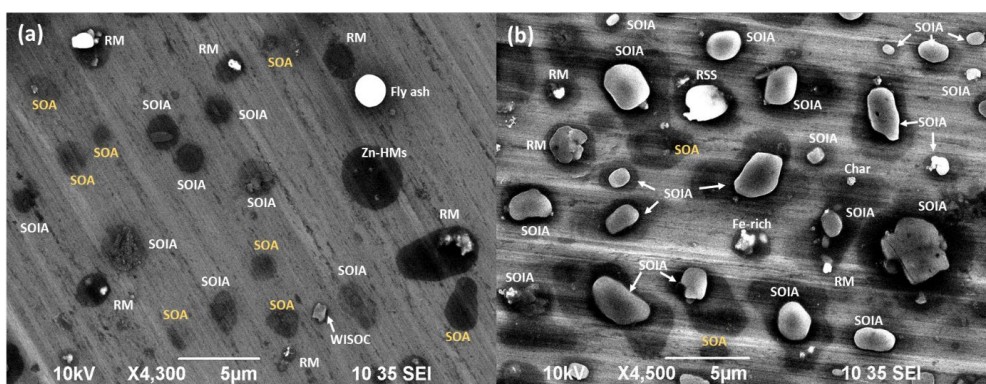

**Figure 5. Typical secondary electron images of $PM_{1-2.5}$ aerosol particles collected on (a) 5/23 PM and (b) 5/25 AM. (RM : reacted mineral dust)**



abundance of individual aerosol particles during 5/25-5/28 (Period II) differed significantly
compared to those observed in Period I. In terms of particle morphology, Fig. 5b shows that
SOIA particles on 5/25 exhibited a bright crystalline morphology, suggesting that these
particles are primarily composed of inorganic components such as sulfate, nitrate, and
ammonium, as described in Section 3.2.1. These bright crystalline SOIA particles mostly
contain high sulfur contents, as shown in Fig. 1b, suggesting that their major composition is
likely ammonium sulfate (AS) (**Wu et al., 2019**). Ammonium-rich conditions in East Asia
facilitate the existence of secondary particles in AS or AN forms (**Kim et al., 2020; Kim et al.,**
**2021**). The ratio of SOIA particles to total particles increased dramatically during Period II, as
shown in Fig. 6. In the $PM_{1-2.5}$ fraction, the proportion of SOIA particles out of the total
particles increased significantly to 61.5% on 5/25, compared to the overall average of 24%,
and remained high at 46.3% during Period II. Additionally, the reacted/aged mineral dust and
SSA particles containing sulfate were dominant during Period II both in the $PM_{1-2.5}$ and $PM_{2.5-10}$
fractions. The drastic increase in ambient PM concentration during this period (Figure S3) is
indicative of an air pollution (haze) episode. In contrast to Period I, which is considered to be
influenced mainly by local emissions due to air mass stagnation, the drastic increase in sulfate
composition of secondary aerosols, reacted mineral dust, and reacted SSA particles during

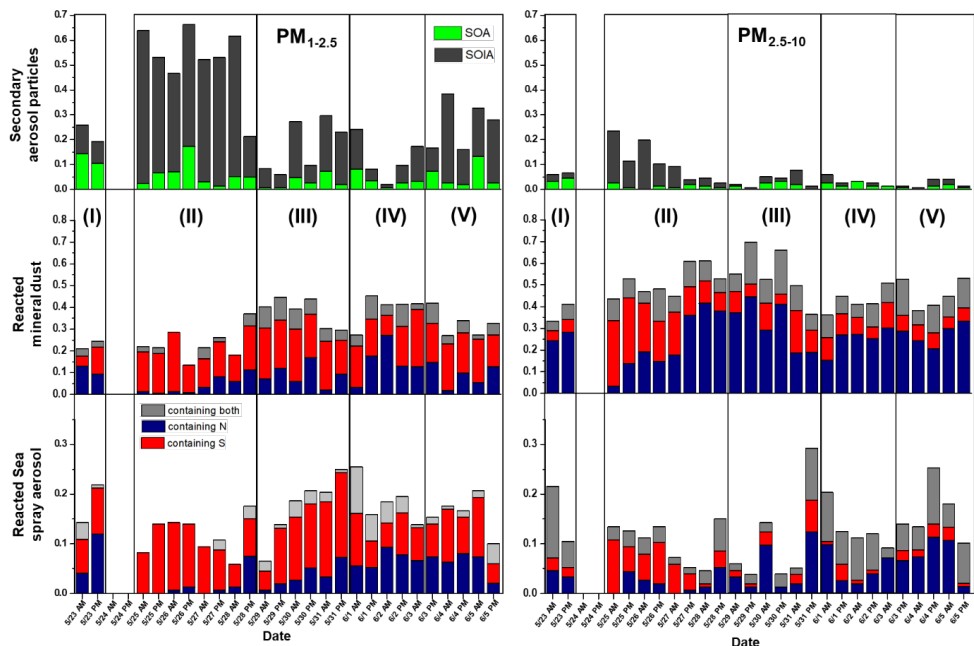

**Figure 6. Relative abundances of secondary aerosols, reacted mineral dust, and reacted SSA particles during the KORUS-AQ campaign.**





Period II seems to be driven by other external factors than local emissions. As shown in Fig.
S4b, air mass transportation from northeastern China at low altitudes (250 m A.G.L) was
observed during this haze episode, which contrasts with Period I. Mild southwesterly winds
(<5 m/s) facilitate the transport of pollutants from China to the study region **(Peterson et al.,**
**2019; Nault et al., 2018; Heim et al., 2020; Choi et al., 2019).** An elevated level of secondary
inorganic constituents, including sulfate, nitrate, and ammonium, was also reported during this
period (**Kim et al., 2018a; Kim et al., 2018b; Song et al., 2022**). The humid conditions (RH
> 60%) sustained during this period provided a favorable environment for the formation of
secondary particles (SIA and SOA) (**Peterson et al., 2019**). Also, sulfate-containing mineral
dust and SSA particles were abundantly observed because the air masses were buffered rapidly
with sulfate when they passed through urban and industrial areas during long-range
transportation (**Yu et al., 2020**). Overall, the data from Period II suggest a significant influence
of long-range transported air masses from northeastern China on the composition and
concentration of aerosol particles in the study region.

**Period III (5/29-5/31)**

The relative abundances of individual particles during 5/29-5/31 (Period III) differed

from those of 5/25-5/28 (Period II), despite consistently high PM concentrations during the
period (Fig. S3) and air mass flow from Northeastern China (Figs. S4c and S4d). During Period
III, the proportion of SOIA particles decreased to 14.4% compared to 46.3% in Period II (Fig.
6). Concurrently, the proportions of reacted mineral dust and SSA particles increased with a
noticeable increase in nitrate-containing ones (Fig. 6). The increase in nitrate-containing
particles in the urban area suggests a strong influence of local emissions (**Yan et al., 2015**).
Changes in the relative abundances of individual HM particles were also noticeable (Fig. 7).
The proportion of Zn-HMs increased rapidly from 0.8% during Period II to 2.8% during Period
III, suggesting an elevated influence of local emissions, given that the major sources of Zn-
HMs are local emissions (section 3.1.5). The proportion of other HMs also somewhat increased
to 2.8% compared to the overall average of 1.9%. The changes in the relative abundances of
individual particles observed in this study are fairly different from other bulk studies in which
air pollution episodes with consistently high inorganic contents were observed during 5/25-
5/31 (**Kim et al., 2018a; Kim et al., 2018b**). Similar to Period II, weak westerly winds
facilitated the transport of pollutants during Period III; however, the formation of secondary
particles appears to be relatively reduced due to ~20% lower RH compared to Period II
(**Peterson et al., 2019**). Additionally, as shown in Figs. S4b and S4c, during Period III, the




travel distance and residence time within the Korean Peninsula were longer relative to Period
II, suggesting an increased mixing of transported and local pollutants. Based on the decrease
in the proportion of secondary aerosol particles formed mainly through gas-particle conversion
and the increase in the reacted forms of the primary aerosol particles, including mineral dust
and SSAs, it is plausible that aggregation or mixing between individual particles intensified
during Period III. Overall, the changes in particle abundances during Period III indicate a
complex interplay of local emissions and long-range transport. The decrease in secondary
aerosol particles and the increase in reacted primary aerosol particles, along with the elevated
proportions of Zn-HMs and other HMs, suggest intensified mixing of pollutants from various
sources during this period.

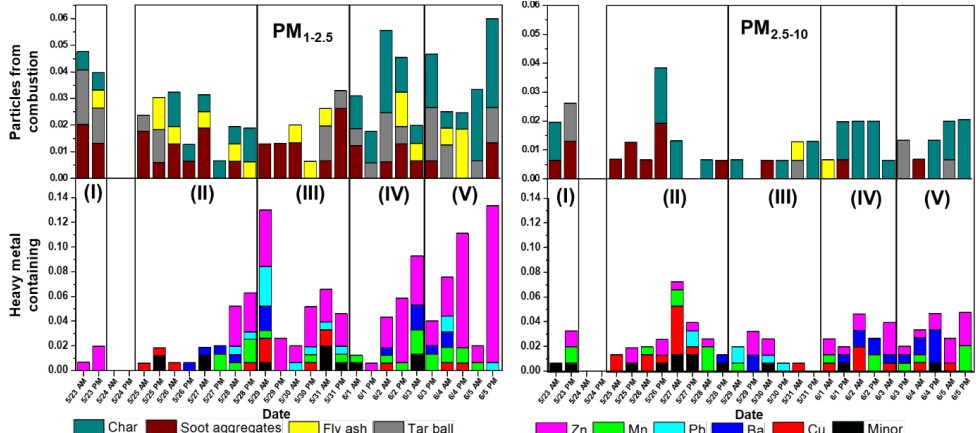

**Figure 7. Relative abundances of combustion particles and heavy metal-containing particles during the KORUS-AQ campaign.**

**Period IV (6/1-6/3)**

After a series of air pollution episodes, ambient PM concentrations decreased
drastically from 6/1 (Fig. S3). The relative abundance of individual particle types during 6/1-
6/3 manifested distinct differences compared to Periods I-III. During Period IV, the proportion
of secondary aerosol particles decreased to 12.9%, compared to an overall average of 29.2%,
while the proportion of genuine mineral dust particles increased from an overall average of
12.9% to 18.3% (Fig. 4). An increase in nitrate-containing reacted mineral dust and SSA
particles was observed in both size fractions (Fig. 6). Zn-HMs from local emissions were also
frequently encountered during this period (Fig. 7). Moreover, an increase in tar balls and char
particles was observed (Fig. 7). Increases in nitrate-containing particles, Zn-HMs, and



combustion particles suggest an intensified influence of local emissions. Additionally, it was
reported that a blocking pattern influenced by high atmospheric pressure was observed over
East Asia during this period, which minimized the transportation of pollutants from other Asian
mainland areas (**Heim et al., 2020; Peterson et al., 2019**). This blocking pattern could have
contributed to the increased influence of local emissions and the observed rise in nitrate-
containing particles, Zn-HMs, and combustion particles during Period IV. Overall, the data
from Period IV suggest that local emissions played a dominant role in shaping the aerosol
composition during this period, with limited influence from long-range transported air masses.
The decrease in secondary aerosol particles and the increase in genuine mineral dust particles
and locally emitted pollutants such as Zn-HMs and combustion particles further support this
conclusion.
**Period V (6/4-6/5)**
Ambient PM concentrations slightly increased during Period V compared to Period
IV (Fig. S3). There was a noticeable increase in Zn-HMs during this period, with the proportion
of Zn-HM increasing from an average of 2.6% to 6.6%. This increase was particularly
noticeable during the afternoon hours (Fig. 7) and might be related to heightened weekend
traffic, as the sampling area is a park surrounded by thoroughfares (Fig. S1). In addition, the
proportion of secondary aerosol particles increased drastically to 28.8% compared to 12.9% in
Period IV, and the proportion of combustion particles somewhat increased to 3.6% compared
to an average of 2.6%. Air mass trajectories shown in Fig. S4f suggest an intensified influence
from inland Korea during Period V, resulting in the increased levels of particles from local
emissions and secondary aerosol particles. When the air mass travels a short distance (~250
km/day), urban areas could be primarily influenced by local emissions (**Lee et al., 2019a**).
Continued blocking patterns from Period IV effectively exclude pollutant transport from
outside areas, but occasionally lead to stagnant conditions (**Peterson et al., 2019**). Overall, the
changes in particle abundances during Period V indicate an intensified influence of local
emissions and secondary aerosol particles, likely due to weekend traffic and stagnant
conditions. The increase in Zn-HMs and combustion particles further supports the impact of
local anthropogenic emissions on air quality during this period. Efforts to manage and control
local emission sources, including vehicle emissions, waste incineration, and fossil fuel
combustions, could play a crucial role in improving air quality in the urban area.
**4. Conclusions**



Individual aerosol particles collected at Olympic Park, Seoul, Korea, during the
KORUS-AQ campaign were analyzed using low-$Z$ particle EPMA. A total of 8004 particles
from 52 samples were examined to identify their chemical species, particle-particle variability,
sources, and atmospheric fate. The major constituents in the $PM_{2.5-10}$ and $PM_{1-2.5}$ fractions were
mineral dust, SSAs, and secondary aerosol particles. However, the relative abundance of
individual particle types varied depending on changes in air mass flow and differences in
emission sources. The reacted mineral dust and reacted SSA particles containing nitrate were
abundant in the $PM_{2.5-10}$ fraction, whereas sulfate-containing ones were relatively higher in the
$PM_{1-2.5}$ fraction. Of particular interest, heavy metals were found to account for a relatively high
proportion of particles both in the $PM_{2.5-10}$ (2.65%) and $PM_{1-2.5}$ (4.42%) fractions, with Zn, Pb,
Ba, Mn, and Cu being the major species. Zn and Pb are mainly released from sources such as
waste incineration, vehicle exhaust, and coal-fired power plants, while Mn, Ba, and Cu are
primarily released from mining and metal alloy industries.
The relative abundances of secondary aerosol particles varied significantly during the
sampling period, reflecting changes in air mass stagnation and emission sources. During the
haze episodes, sulfate-containing particles, including SOIA, mineral dust, and SSAs, were
predominant, and the proportion of SOA particles increased as local influence intensified.
During the clean period of 6/1-6/3, nitrate-containing particles were abundantly observed,
indicating a high contribution of $NO_x$ emissions from local sources. Zn-HMs from local sources
such as vehicle emissions and waste incineration were noticeably observed during 6/4-6/5
when the air mass stagnated over the Korean peninsula.
The temporal variations in the abundance and physicochemical characteristics of
individual aerosol particles provide valuable insights into the behavior and emission sources
of atmospheric urban aerosols. The changes in the composition of organic and inorganic
components resulted in distinct morphological and crystalline structures of secondary aerosol
particles, influencing properties such as hygroscopic behavior and radiative forcing. The
relative abundance of HMs, particularly those containing Zn, effectively reveals the impact of
local emissions such as vehicle emissions and waste incineration. The highly hygroscopic
nature of the observed Zn-HMs suggests a potential threat to human health, as they are prone
to adsorbing or reacting with other organic and inorganic components in the atmosphere. The
observed changes in the abundance of particles from typical combustion events and secondary
aerosol particles emphasize the need to manage local emission sources to maintain air quality.
The complexity of aerosol particle behavior highlights the importance of a comprehensive



understanding of the interplay between local emissions, long-range transport, and
meteorological conditions to develop effective air pollution mitigation strategies.

**Code and data availability**
The data set is available upon request from Chul-Un Ro (curo@inha.ac.kr).

**Author contributions**
**Chul-Un Ro (RCU)** and **Geng Hong (GH)** designed and supervised the entire experimental
program, provided guidance on the quantitative analysis of individual particles, and reviewed
the manuscript. **Hanjin Yoo (HJY)** conducted the single-particle analysis, analyzed the data,
and wrote the manuscript. **Li Wu (LW)** reviewed the manuscript and provided feedback on
the manuscript. All authors read and approved the final manuscript.

**Competing interests**
The authors declare that they have no conflict of interest.

**Financial Support**
This study was supported by the National Research Foundation of Korea (NRF) grant funded
by the Korean government (MSIT) (No. 2021R1A4A1032579 and No. 2021R1A2C2004240)
and by the National Institute of Environmental Research (NIER) funded by the Ministry of
Environment (MOE) of Korea (NIER-2021-03-03-007). The authors thank for a fund from the
China State High-end Foreign Expert Recruitment Project (G2022004013L).

**Supporting Information**
Table S1 and Figures S1-S4

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
