# Peer review of "Physicochemical and Temporal Characteristics of Individual Atmospheric Aerosol Particles in Urban Seoul during KORUS-AQ Campaign: Insights from Single-Particle Analysis"

_EGUsphere, 2023_

## Author Response (AR1)

**Comments (*in italics*) and responses**

**Reviewer #1**

*Yoo et al. present a concise and well-written investigation of individual particles collected in Seoul as part of the KORUS-AQ campaign, with the goal of investigating the different particle types that might be found as a result of different atmospheric events. The authors furthermore specifically highlight the composition of particles in the supermicron size range. Some of the core findings aren't particularly novel (e.g., from the abstract: "atmospheric aerosol particles in urban area vary depending on their size, sources, and reaction or ageing status.."), and have been well known to the atmospheric community at large for some time. However, the authors don't attempt to sell this as the novel aspect, and instead emphasize the need to establish these kinds of relationships for particle types that aren't generally well characterized in a variety of environments (e.g., supermicron). The relatively high abundance of heavy metal particles that were observed is particularly interesting as well. I believe that this study will be publishable in ACP pending the addressment of several minor concerns listed below:*

**Response**: Thank the reviewer very much for his/her positive evaluation and constructive feedback on our work. We appreciate the reviewer's valuable suggestions and are glad that the reviewer found our work insightful and worthy of publication. In response to the reviewer's comments, we have made the necessary modification to our paper.

*Page 4 Lines 120 – 123: Please use the HYSPLIT references that NOAA specifically requests (Stein et al; Rolph et al; https://www.ready.noaa.gov/HYSPLIT_traj.php).*

**Response**: Thank you for your comment. We added that references (p 4, lines 124-125 in the marked revised version)

*Page 4 Lines 126 – 128: The vacuum conditions needed to analyze particles via SEM should be acknowledged, and caveats should be listed for how this affects the final particle characterization (e.g., bias against more volatile components).*

**Response**: As the reviewer mentioned, measurements under the vacuum condition can lead to evaporation of volatile components. However, this impact is likely to be insignificant considering the general composition of ambient supermicron aerosols. We modified corresponding section respecting the reviewer's comment: (p 4, line 130 and p5, lines 141-144 in the marked revised version): 'Measurements under the vacuum condition may result in the evaporation of volatile organic components in individual aerosol particles, but these effects are negligible for ambient supermicron aerosols given their general chemical compositions.'

*Page 5 Line 149: The referred to particle classification scheme in the SI is somewhat unclear – the particle classification descriptions appear rather qualitative at times (e.g., 'particles containing many heavy metals') rather than a more precise, quantitative description (e.g., what quantitative abundance of which specific heavy metals do the authors use to assess a particle as one containing heavy metals?). As a result, it is somewhat difficult to get a true appreciation for how particle classification was done.*

*Even if the authors are reproducing a previously published classification scheme, the precise information should be reproduced in the SI and if necessary, referenced accordingly.*

**Response**: We modified our description for particle classification to ensure that they are both quantitative and specific (Please see newly modified Supporting Information Section A and Table S1 in the marked revised version)

*Figure 3-4: Consider using a different color scheme that is more amenable to color-blindness.*

**Response**: We changed Figures 3, 4, 6 and 7 to include more appropriate color schemes and patterns to ensure readability for all users.

*Page 11 Lines 313 – 314: While the authors suppositions here may be true, they should caveat them by noting the extremely number of total particles that lead them to these observations.*

**Response**: We agreed with the reviewer's concern and modified the sentence as follows: 'The observation that Mn, Ba, and Cu-HMs appear abundantly as a mixture of Fe or mineral dust in this study suggests a possibility that their major source might be ferroalloy plants, mining, or ore crushing plants.' (p 12, lines 318-320 in the marked revised version)

*Figure 4: Consider labeling what the various stages are in the figure, or alternatively, at least list it in the figure caption.*

**Response**: We changed the labels in Figure 4 to make various stages clearer.

*Page 14 Lines 369 – 370: Have the authors investigated the potential of biomass burning/wildfires specifically instead of just combustion in general? Consider the NASA fire map resource.*
**Response**: Thank you for your comment. We modified the sentence acknowledging the possible additional influence of wildfires from Siberia region during this period (pp 14-15, lines 374-378 in the marked revised version): 'The enhanced level of combustion particles suggests the contribution of local emissions, while it also correlates with previous studies (Song et al., 2022; Peterson et al., 2019) that indicate additional influences from Siberian wildfires between 5/20-5/23.'

**Reviewer #2**

*This study investigated physicochemical properties of individual particles during the KORUS-AQ campaign using a quantitative electron probe X-ray microanalysis. This study classified these individual particles into seven types and atmospheric situations into five types. This study showed relative abundances of different types of particles during five types of atmospheric situations. This study found that Zn-containing particles were mostly sourced from local vehicle exhausts and waste incinerations, while Mn, Ba, and Cu-containing particles were mainly attributed to metal-alloy plants or mining emissions. This study suggested that the morphology and chemical compositions of particles*

*were affected by their sources and ageing processes. Although this study clearly expressed the results, some problems listed below need to be addressed before the manuscript could be published.*

**Response**: Thank the reviewer for the reviewer's positive evaluation and comprehensive feedback on our study. We are grateful for the reviewer's understanding of our objectives and findings. We acknowledge the concerns the reviewer pointed out and are committed to carefully resolving each issue to improve the quality and clarity of our paper.

*L16: Please indicate the full name of 'KORUS-AQ'.*

**Response**: Thank you for your comment. We indicated 'Korea-US Air Quality' (p 1, line 16 in the marked revised version)

*L117: The second sampling time (15:00 ~ 16:00) should be afternoon. I don't think it's appropriate for the authors to consider this period as an evening.*

**Response**: We changed the evening to the afternoon (p 4, line 118 in the marked revised version)

*L117-118: What is the range of sampling duration? The authors should show it.*

**Response**: We added the details about the sampling duration (p 4, line 120 in the marked revised version): '10-30 minutes for stage 2 and 5-15 minutes for stage 3'

*Section 3.1: The authors classified individual particles into seven types, but most of these seven types of particles are externally mixed. As we know, particles are usually internally mixed in aged air, especially during polluted periods. Many researchers have found this phenomenon in urban air using high-resolution TEM, such as internal mixing of carbonaceous and secondary inorganic aerosols (Adachi & Buseck, 2013; Li et al., 2021; Zhang et al., 2022). I suggest the authors to further consider the internal mixing of secondary aerosols and other particles (e.g., carbonaceous or metal-containing particles) to better evaluate the source of particles.*

**Response**: Thank you for your valuable comment. We agree that individual particles, particularly in the polluted air, can be characterized more in detail by investigating their internal mixing, and we are grateful for the references the reviewer provided. The significance of internally mixed particles in pinpointing pollution sources and understanding their health and climate implications is well recognized by us, too. However, given the extensive dataset of over 8,000 particles in this study, our primary objective was to delve deeply into the physicochemical and temporal characteristics of these particles, relating them to atmospheric conditions and transport pathways. To reflect the importance of internal mixing and to clarify our intent, we modified Section 3.1, ensuring the inclusion of reviewer's valuable comments and suggested references (p 5 lines 154-158 in the marked revised version): 'While the internal mixing state of individual aerosol particles can offer valuable insights into the sources and formation mechanisms of ambient aerosols (Adachi and Buseck., 2013; Li et al., 2021; Zhang et al.,

2022), this study primarily focuses on the overall physicochemical characteristics and relative abundances of the ambient aerosols due to the extensive number of particles investigated.'

*Some data sources did not be indicated, such as '73.2% and 44.5%' on L181, '71.0%' on L189, and etc. A table may be appropriate.*

> **Response**: We changed Table 1 to include those numbers.

*L272: 'As' is not a heavy metal element. Correspondingly, some results in section 3.1.5 and figures (e.g., Figure 3) need to be changed.*

> **Response**: The term "heavy metal" has been somewhat ambiguously used, and its definition can vary based on the context. As the reviewer pointed out, 'As (arsenic)' is, by strict chemical classification, a metalloid, which means it possesses properties intermediate between metals and nonmetals. Nonetheless, the term "heavy metal" in environmental and toxicological contexts often encompasses both toxic metals and certain metalloids, like arsenic. This classification is informed not only by the element's strict chemical classification but also by its practical implications related to density and toxicity. Specifically, arsenic has a significant density (5.7 g/cm$^3$) and notable toxicological effects on organisms, as highlighted in studies such as Tian et al., 2015 (in the manuscript), and following references.

Nies. D. H., Microbial heavy-metal resistance, Appl. Microbiol. Biotechnol., 51, 730-750, 1999.
Li et al., The preferential accumulation of heavy metals in different tissues following frequent respiratory exposure to PM2.5 in rats, Sci. Rep., 2015, DOI: 10.1038/srep16936.
Mc Neill et al., Large global variations in measured airborne metal concentrations driven by anthropogenic sources, Sci. Rep., 2020, doi.org/10.1038/s41598-020-78789-y.L340:

*During period I, the air mass is more likely to be long-range transported based on backward trajectories in Figure S4a. How did the authors determine the period I as a local polluted event?*

> **Response**: As shown in Fig. S4a, the majority of the airmass on that day came mainly via Korea's inland regions, indicating a strong influence of local emissions. The PM concentration shown in Fig. S3 also exhibit relatively higher value than clean period (6/1-6/3). Moreover, the ratio of SOA to secondary aerosol particles was observed to be approximately 2.5 times higher on that day. This observation aligns with findings from Nault et al. (2019), Kim et al. (2018a), and Kim et al. (2018b), which indicated that the primary sources of SOA in Korea are local emissions. Studies from the KORUS-AQ campaign, conducted during the same timeframe, further suggest that air masses experienced stagnation under persistent high pressure. This stagnation likely resulted in local emissions becoming the predominant contributors (Kim et al., 2018b; Peterson et al., 2019; Heim et al., 2020).

*L375-384: The sentence is repeated.*

> **Response**: Thank you for your kind pointing out. We edited the repeated sentences.

*L398-399: This result confuses me. Based on backward trajectories in Figure S4b, air masses at high altitudes (1000 m A.G.L) should be transported from northeastern China rather than that at low altitudes showed by the authors. Air masses at low altitudes are mainly from the local area.*

**Response**: The reviewer correctly observed that the air mass at a high altitude (1000 m A.G.L) displayed in Fig. S4b originates from northeastern China. Nevertheless, the air masses at lower altitudes (250 m and 500 m A.G.L) directly reached Korea via the Shandong Peninsula and the Yellow Sea. The duration for which low-altitude air masses spent over the Korean inland before reaching the sampling site was shorter than 3 hours. The spatial coverage and transit time of the air masses over Korean terrain shown in Fig. S4b were considerably smaller and shorter compared to their journey during periods of local pollution as shown in Fig. S4a.